# Effects of breathing exercise and thoracic techniques on pain and disability in low back pain: A systematic review and meta-analysis

Tahere Seyedhoseinpoor[1]*, Ramin Jafari[2], Zohreh Shafizadegan[2],
Maryam Abbaszadeh-Amirdehi[1]

**1** Mobility Impairment Research Center, Health Research Institute, Babol University of Medical Sciences, Babol, I.R.Iran, **2** Musculoskeletal Research Center, Department of Physical Therapy, School of Rehabilitation Sciences, Isfahan University of Medical Sciences, Isfahan, Iran

* tahere.hoseinpoor.65@gmail.com

## Abstract

### Purpose

The objective of this study was to systematically review the effectiveness of thoracic-focused interventions, including breathing exercises and thoracic manual techniques (mobilization, high-velocity low-amplitude manipulation, and release techniques), on pain and disability in patients with low back pain (LBP).

### Methods

PubMed, Scopus, Web of Sciences, ProQuest, Ovid, Physiotherapy Evidence Database (PEDro), Cochrane Central Register of Controlled Clinical Trials (CENTRAL), and Google Scholar were searched without language restrictions. Clinical trials with control groups on pain and disability in low back pain patients focusing on the efficacy of breathing exercises or thoracic technique were included. In total, 31 studies contributed to the meta-analysis for pain and 24 for disability.

### Results

Pooled analyses using Morris' dppc demonstrated a statistically significant, small effect for pain reduction (dppc = −0.35, 95% CI = −0.46 to −0.23) and a large effect for disability improvement (dppc = −0.71, 95% CI = −0.86 to −0.57) when compared with control groups. Thoracic manual techniques showed larger effects on both pain and disability compare to breathing exercises. However, substantial statistical heterogeneity (I² > 85%) persisted in most analyses.

**Data availability statement:** All relevant data are within the paper and its Supporting information files.

**Funding:** The author(s) received no specific funding for this work.

**Competing interests:** The authors have declared that no competing interests exist.

## Conclusion

Breathing and thoracic manual techniques may be effective in reducing disability and, to a lesser extent, pain in patients with LBP, but the overall certainty of evidence is low. However, the quality of the evidence is low. Variability in treatment protocols, study quality, blinding, and outcome measures likely contributed to inconsistencies. Further high-quality trials with standardized protocols are needed to confirm these findings and inform clinical practice.

## Introduction

Chronic low back pain (CLBP) is one of the most common musculoskeletal conditions affecting most of the population [1]. The incidence of CLBP is associated with substantial medical costs and a significant burden on families and society [2,3], and it is the leading cause of years lived with disability [1]. Multiple factors, including biopsychosocial elements, contribute to the development of CLBP, although it may occur in the absence of any discrete pathology. In addition, joints, intervertebral discs, tendons, ligaments, and muscles may be separately involved in its development [4]. Research shows a significant association between persistent or recurrent low back pain (LBP) and deep trunk muscle weakness, especially core muscles, as well as poor coordination and trunk proprioception [5–7].

Although CLBP can usually be treated with traditional physical therapy or manual therapy, most patients do not fully recover from their symptoms. Recently, specialists have emphasized the correlation between LBP and respiratory disorders [8,9]. The respiratory function and spinal stability are supported by the core muscles and the diaphragm. Respiratory dysfunction impairs diaphragm function and, consequently, the diaphragm's role in spinal stability [10]. Beeckmans et al found that immune, biomechanical, psychosocial and socioeconomic factors may explain the association between CLBP and respiratory disease [8]. Ostwal et al. found that over 71% of individuals suffering from chronic back pain have deviant breathing patterns during movement control tests and irregular movement patterns at rest [11].

Breathing exercises and thoracic manual techniques effectiveness in CLBP management lies in their shared neuromuscular and biomechanical mechanisms. The diaphragm, as a dual-function muscle for respiration and postural control, modulates intra-abdominal pressure (IAP) and lumbar stability through its synergistic action with deep core muscles (transversus abdominis, multifidus) and the thoracolumbar fascia [12]. In CLBP, aberrant breathing patterns (e.g., upper chest dominance) disrupt this synergy, leading to diaphragm dysfunction, reduced IAP, and compensatory lumbar overload [11]. Concurrently, thoracic spine stiffness or ribcage hypomobility -common in CLBP- further impairs diaphragmatic excursion and perpetuates faulty movement patterns [13]. Thoracic mobilization techniques address these restrictions by restoring joint mobility and fascial continuity, while targeted breathing exercises re-establish diaphragmatic-postural coordination. Together, these interventions break the cycle of respiratory dysfunction and spinal instability: thoracic techniques create

a mechanically optimal environment for diaphragmatic engagement, and respiratory training reinforces this adaptation through motor relearning [14].

Despite the potential benefits of breathing exercises and trunk manual techniques, evidence for their effectiveness in patients with LBP is still emerging. So far, there have been previous systematic reviews that have analyzed the efficacy of breathing exercises in individuals with LBP [15–18], and additionally, two other systematic reviews have investigated the impact of manual treatment on musculoskeletal and respiratory factors in LBP [19,20]. However, no systematic review or meta-analysis has yet evaluated the combined effectiveness of breathing exercises and thoracic techniques in CLBP. Supporting evidence from other musculoskeletal conditions further strengthens the rationale for this work. Several studies have demonstrated that breathing exercises improve pain, function, and quality of life in chronic neck pain [21], shoulder dysfunction [22], and postural syndromes such as thoracic kyphosis and forward head posture [23,24]. These findings suggest that breathing interventions exert musculoskeletal benefits beyond LBP, likely through their influence on neuro-muscular coordination, posture, and pain modulation.

Accordingly, the objective of this systematic review was to synthesize current evidence on the effectiveness of thoracic-focused interventions – including breathing exercises and thoracic mobilization/release techniques – on pain and disability in patients with CLBP. By consolidating the available literature, this review aims to provide clearer insight into the clinical utility of these interventions and to identify priorities for future research.

## Methods

The systematic review and meta-analysis was based on PRISMA (Preferred Reporting Items for Systematic Reviews and Meta-Analyses) statement (S1 Appendix) [25]. The initial protocol of this systematic review has been registered prospectively to the International Prospective Register of Systematic Reviews database (PROSPERO; CRD42023439503).

### Search methods for the identification of studies and selection of criteria

Initially, an extensive online database searches were performed in PubMed, Scopus, Web of Sciences, ProQuest, Ovid, Physiotherapy Evidence Database (PEDro), Cochrane Central Register of Controlled Clinical Trials (CENTRAL) and Google Scholar from January 1, 1990, to May 1, 2025, without any language restrictions. The search syntax was developed by combining keywords with Medical Subject Headings (MeSH) terms, where available, in PubMed, and the underlying search strategy was modified to optimize for each of the other databases (Full search syntax provided in S2 Appendix).

To identify additional primary trials and eligible articles, reference lists of included articles and related reviews were scanned, and manual keyword searches of ClinicalTrials.gov and the WHO International Clinical Trial Registry Platform were performed. Authors of published posters or conference proceedings were contacted by e-mail to request further information (if available).

Search results were imported into EndNote software (version X8; Thomson Reuters, NY, USA), and identified duplicates were removed in the software. All titles and abstracts of the remaining studies were preliminarily screened by one reviewer to identify potentially eligible studies. The full text of potentially eligible articles was then assessed by three independent reviewers, and studies were included if they met the following criteria: 1) controlled trials, 2) sample populations consisted primarily of individuals with low back pain, 3) the intervention methods used in these trials focus on the effectiveness of breathing exercises alone or as an adjunct treatment, diaphragmatic exercises, diaphragm activation techniques, or thoracic mobility techniques (thoracic mobilization, high-velocity low-amplitude manipulation, and release techniques), and 4) patient pain or disability scores were reported as the primary or secondary outcome. Studies without a control group, studies in participants with prior lumbar spine surgery or LBP related to fracture, malignancy, infection, or inflammation, and studies measuring the effect of exercise such as wuinxi, tai chi, qigong, yoga, Pilates, running, walking, and relation technique were not included. Consensus was the method used to resolve disagreements that arose during the study selection phase.

### Types of outcomes

Low back pain intensity and disability were the predefined primary outcomes and included studies had to report at least one of the described primary outcomes.

### Data extraction and management

Two independent reviewers extracted the study data using a data extraction form developed for the study. One of the reviewers checked all data entries for missing information or inaccuracies. Disagreements were also resolved by consensus. The extracted data are shown in detail in Table 1 and S1 File.

### Assessment of risk of bias in included studies

The risk of bias among the included studies was assessed by two reviewers independently of each other according to the Cochrane Collaboration's Rob-2, which includes the following domains: risk of bias due to confounding, risk due to the randomization process, risk due to deviation from the intended interventions, risk due to missing outcome data, risk in the measurement of the outcome, risk in the selection of the reported outcome, and risk of overall bias. The terms "low risk of bias", "some concerns", and "high risk of bias" were used to assign a specific rating to each domain. The Risk-of-Bias VIsualization tool (robvis) was used to generate the risk of bias graph (Cochrane and robvis) [60]. Interrater agreement was assessed through Gwet's first-order coefficient of agreement (Gwet's AC1) with its corresponding 95% CI and interpreted using Landis & Koch's rule of thumb (0 to 0.20 for low, 0.21 to 0.40 for fair, 0.41 to 0.60 for moderate, 0.61 to 0.80 for substantial, and 0.81 to 1 for near perfect agreement) [61]. In contrast to Cohen's kappa, Gwet's AC1 is robust against a "kappa paradox" in which low kappa values are reported despite observed agreement values [62]. Disagreements were also the subject of consensus.

### Measures of treatment effect

Low back pain intensity and disability were expressed as Morris's delta (Morris's dppc) with a 95% confidence interval (CI) to estimate the magnitude of the treatment effect while considering the pre-post value in both treatment and control groups [63,64]. Effect sizes were computed Using the effect size calculator from the Campbell Collaboration (http://ww.campbellcollaboration.org/escalc/html/EffectSizeCalculator-SMD-main.php). Three different categories were used to quantify the magnitude of effect sizes: small (dppc<0.40), medium (0.40 ≤ dppc ≤ 0.70), and large (dppc>0.70). The assessment of a meaningful clinical effect relied on the existence of an effect size that was at least moderately significant [65].

### Assessment and investigation of heterogeneity

The heterogeneity among primary studies was assessed using the $I^2$ statistic and the Q test ($\chi^2$), in accordance with the guidelines provided by the Cochrane Handbook for Systematic Reviews of Interventions [66]. The $I^2$ statistic was interpreted in accordance with the following guidelines: a range of 0–40% indicates no significant heterogeneity, while a range of 30–60% suggests moderate heterogeneity. A range of 50–90% indicates substantial heterogeneity, while a range of 75–100% suggests considerable heterogeneity [67]. Prior to conducting the pooled analysis, the existence of heterogeneity was taken into account. When the $I^2$ values above 50% and the confidence intervals (CIs) overlapped upon visual examination of the forest plot, the findings were pooled in a meta-analysis utilizing a random effects model [68]. Statistical significance was determined by a p-value less than 0.05.

In the subgroup analysis, the meta-analysis was performed independently for the subgroups of age, patient blinding, rater blinding, quality (risk of bias), sample size, treatment characteristics, treatment duration, treatment sessions, and therapist supervision in two separate meta-analyses including 24 and 31 studies, respectively. Meta-regression was not conducted because the number of studies per covariate was insufficient to produce reliable estimates, and several moderators were categorical. This approach is consistent with PRISMA guidelines for meta-analyses with a modest number of studies [69].

**Table 1. Studies are in chronological order from oldest to most recent by year of publication.**

| | Author-year | Study design | Treatment group characteristics | Treatment duration & Type | Control group characteristics | Control duration & Type | Blinding | | Outcome measure |
|---|---|---|---|---|---|---|---|---|---|
| | | | | | | | Patient | Assessor | |
| 1 | Mehling 2005 [26] | Multi-center RCT | N = 14 CLBP Age: 49.7 ± 12.1 Male: 5 Female: 11 | • 6 weeks • 12 treatment sessions • Breath therapy | N = 12 CLBP Age: 48.7 ± 12.5 Male: 5 Female: 7 | • 6 weeks • 12 treatment sessions • Physical therapy | no | no | Pain: VAS Disability: RMQ |
| 2 | de Olive-ria 2013 [27] | Sing-center Single-blind RCT | N = 74 NCLBP Age: 45.95 ± 12.30 Male: 24 Female: 50 | • Single session • High velocity manipulation T1-T5 | N = 74 NCLBP Age: 46.32 ± 10.22 Male: 15 Female: 59 | • Single session • High velocity manipulation specific to the painful lumbar vertebrae (L2 to L5) | no | yes | Pain: NRS Disability: - |
| 3 | Heo 2015 [28] | RCT | N = 12 CLBP Age: 43 ± 5.4 Male: 8 Female: 4 | • 12 weeks • 36 treatment sessions • Physical therapy & Thoracic mobilization &Exs | N = 12 CLBP Age: 45.4 ± 6.4 Male: 11 Female: 1 | • 12 weeks • 36 treatment sessions Physical therapy & Exs | no | no | Pain: VAS Disability: - |
| 4 | Ho-Hee 2015 [29] | RCT | N = 9 CLBP Age: 23.31 ± 4.55 Male: N/A Female: N/A | • 8 weeks • 40 treatment sessions • Abdominal breathing exs & Stabilization exs | N = 9 CLBP Age: 22 ± 0.86 Male: N/A Female: N/A | • 8 weeks • 40 treatment sessions • Stabilization exs | no | no | Pain: - Disability: ODI |
| 5 | Babina 2016 [13] | Single-blind RCT | N = 31 NCLBP Age: 40.75 Male: N/A Female: N/A | • 2 weeks • 10 sessions • Physical therapy & breath therapy & thoracic mobilization | N = 31 NCLBP Age: 40.75 Male: N/A Female: N/A | • 2 weeks • 10 sessions • Physical therapy & breath therapy | no | yes | Pain: - Disability: ODI |
| 6 | Kang 2016 [30] | Sing-center RCT | N = 10 CLBP Age: 42.5 ± 5.3 Male: N/A Female: N/A | • 6 weeks • 24 sessions • Exhalation exs | N = 10 CLBP Age: 40.1 ± 5.3 Male: N/A Female: N/A | • 6 weeks • 24 sessions • Trunk stabilization | no | no | Pain: - Disability: ODI |
| 7 | Mohanty 2016 [31] | RCT | N = 100 CLBP + spon-dylolisthesis Age: 42.42 ± 6.96 Male: 34 Female: 66 | • 4 weeks • 20 sessions • Physical therapy & • Upper thoracic mobilization | N = 100 CLBP + spondy-lolisthesis Age: 41.76 ± 5.12 Male: 31 Female: 69 | • 4 weeks • Physical therapy | no | no | Pain: - Disability: ODI |
| 8 | Finta 2018 [32] | RCT | N = 26 NCLBP Age: 22.31 ± 5.15 Male: N/A Female: N/A | • 8 weeks • 16 sessions • Complex therapy & Dia-phragm training | N = 21 NCLBP Age: 21.33 ± 4.73 Male: N/A Female: N/A | • 8 weeks • 16 sessions • Complex therapy | no | no | Pain: VAS Disability: - |
| 9 | Marti-Salvador 2018 [33] | Multi-center Double-blind RCT | N = 33 NCLBP Age: 43.4 ± 10.8 Male: 16 Female: 17 | • 4 weeks • 5 sessions • Diaphragm training | N = 33 NCLBP Age: 41.7 ± 10.3 Male: 13 Female: 20 | • 4 weeks • 5 sessions • Sham diaphragm training | yes | yes | Pain: VAS Disability: ODI |
| 10 | Park Ju-jung 2018 [34] | Single-blind RCT | N = 15 CLBP Age: 31.0 ± 16.6 Male: N/A Female: N/A | • Thoracic mobilization (T1-T8) | N = 15 CLBP Age: 32.0 ± 13.7 Male: N/A Female: N/A | • Placebo thoracic mobiliza-tion (T1-T8) | yes | no | Pain: VAS Disability: - |

*(Continued)*

**Table 1.** (Continued)

| | Author -year | Study design | Treatment group characteristics | Treatment duration & Type | Control group characteristics | Control duration & Type | Blinding | | Outcome measure |
|---|---|---|---|---|---|---|---|---|---|
| | | | | | | | Patient | Assessor | |
| 11 | Fan 2018 [35] | Sing-center RCT | N = 30 NLBP Age: 40.87 ± 9.56 Male: 17 Female: 13 | • 4 weeks • 20 treatment sessions • Stabilization exs & Breathing exs | N = 30 NLBP Age: 38.53 ± 11.19 Male: 15 Female: 15 | • Stabilization exs | no | no | Pain: VAS Disability: ODI |
| 12 | Fisher 2019 [36] | Sing-center Double-blind RCT | N = 52 LBP Age: 38.46 ± 12.07 Male: 20 Female: 32 | • 1.5 weeks • 3 treatment sessions • Stabilization exs & Education & thoracic manipulation (T6-T12) | N = 49 lbp Age: 37.78 ± 10.91 Male: 10 Female: 39 | • 1.5 weeks • 3 treatment sessions • Stabilization exs & Education & Sham thoracic manipulation (T6-T12) | yes | yes | Pain: NRS Disability: RMQ |
| 13 | Cho 2019 [37] | Sing-center RCT | N = 15 CLBP Age: 40.93 ± 6.45 Male: 6 Female: 9 | • 12 weeks • 36 treatment sessions • Breathing exs | N = 15 CLBP Age: 42.40 ± 6.93 Male: 6 Female: 9 | • 12 weeks • 36 treatment sessions • Stabilization exs | no | no | Pain: VAS Disability: - |
| 14 | Park Sam-Ho 2019 [38] | RCT | N = 20 LBP Age: 30.9 ± 4.53 Male: 12 Female: 8 | • 4 weeks • 12 sessions • Stabilization exs & respiratory resistance training | N = 23 LBP Age: 30.7 ± 4.32 Male: 12 Female: 11 | • 4 weeks • 12 sessions • Stabilization exs | no | no | Pain: NRS Disability: ODI |
| 15 | Ahmad-nezhad 2020 [4] | Multi-center Single-blind RCT | N = 24 NCLBP Age: 21.43 ± 2.15 Male: 12 Female: 12 | • 8 weeks • inspiratory muscle training & regular sport exs | N = 24 NCLBP Age: 22.33 ± 1.41 Male: 12 Female: 12 | • 8 weeks • Supervised exs • regular sport exs | no | yes | Pain: VAS Disability: - |
| 16 | de Olive-ria 2020 [39] | Sing-center Single-blind RCT | N = 74 NCLBP Age: 45 ± 14 Male: 17 Female: 57 | • 4 weeks • 10 treatment sessions • Thoracic manipulation (T5-T6) | N = 74 NCLBP Age: 45 ± 13 Male: 16 Female: 58 | • 4 weeks • 10 treatment sessions • Mobilization specific to the painful lumbar vertebrae | no | yes | Pain: NRS Disability: RMQ |
| 17 | Lim 2020 [40] | Sing-center Single-blind RCT | N = 12 NCLBP Age: 43.33 ± 10.26 Male: 4 Female: 8 | • Breathing exs | N = 12 NCLBP Age: 45.33 ± 10.27 Male: 4 Female: 8 | • Mobilization specific to the painful lumbar vertebrae | yes | yes | Pain: VAS Disability: - |
| 18 | Divya 2020 [41] | RCT | N = 15 CLBP Age: 38.53 ± 7.84 Male: N/A Female: N/A | • 4 weeks • 12 treatment sessions • Electrotherapy& Stabilization exs & Thoracic mobilization & strength | N = 15 CLBP Age: 40.33 ± 8.56 Male: N/A Female: N/A | • 4 weeks • 12 treatment sessions • Electrotherapy& Stabilization exs | no | no | Pain: NRS Disability: ODI |
| 19 | Kosto-dinovic 2020 [42] | Single-blind RCT | N = 40 CLBP + radicu-lopathy Age: 44.12 ± 10.25 Male: 18 Female: 22 | • 8 weeks • Electrotherapy& Stabilization exs & Thoracic mobilization | N = 40 CLBP + radicu-lopathy Age: 44.3 ± 9.13 Male: 17 Female: 23 | • 8 weeks • Electrotherapy& Stabilization exs | no | yes | Pain: VAS Disability: ODI |

*(Continued)*

| | Author-year | Study design | Treatment group characteristics | Treatment duration & Type | Control group characteristics | Control duration & Type | Blinding | | Outcome measure |
|---|---|---|---|---|---|---|---|---|---|
| | | | | | | | Patient | Assessor | |
| 20 | Oh 2020 [43] | Sing-center Single-blind RCT | N = 22 LBP+instability Age: 46.14 ± 2.59 Male: - Female: 22 | • 4 weeks • 12 treatment sessions • Stabilization exs & respiratory resistance exs | N = 22 LBP+ instability Age: 44.45 ± 2.54 Male: - Female: 22 | • 4 weeks • 12 sessions • Stabilization exs | no | yes | Pain: VAS Disability: ODI |
| 21 | Atilgan 2021 [44] | Sing-center RCT | N = 23 NCLBP Age: 32.08 ± 7.15 Male: - Female: 23 | • 8 weeks • 24 treatment sessions • Stabilization ex & breath therapy | N = 20 NCLBP Age: 37.7 ± 5.80 Male: - Female: 20 | • 8 weeks • 24 sessions • Stabilization exs | no | no | Pain: VAS Disability: - |
| 22 | Otadi 2021 [45] | Multi-center RCT | N = 12 NCLBP Age: 36.2 ± 8.9 Male: 7 Female: 5 | • 4 weeks • 12 treatment sessions • Supervised exs & home exs • Diaphragm training & TENS | N = 12 NCLBP Age: 34.2 ± 10.8 Male: 5 Female: 7 | • 4 weeks • 12 sessions • Supervised exs & home exs • TENS | no | no | Pain: NRS Disability: - |
| 23 | Park Sam-Ho 2021 [46] | Sing-center RCT | N = 20 LBP+ instability Age: 47.09 ± 1.15 Male: - Female: 20 | • 4 weeks • 12 treatment sessions • Stabilization exs & respiratory resistance exs | N = 20 LBP+ instability Age: 46.68 ± 1.73 Male: - Female: 20 | • 4 weeks • 12 sessions • Stabilization exs | no | no | Pain: NRS Disability: ODI |
| 24 | Park Sun-Ju 2021 [47] | Sing-center RCT | N = 22 CLBP Age: 41.09 ± 8.94 Male: 12 Female: 10 | • 8 weeks • 24 treatment sessions • Stabilization exs & Thoracic manipulation | N = 22 CLBP Age: 39.68 ± 9.38 Male: 12 Female: 10 | • 8 weeks • 24 treatment sessions • Stabilization exs | no | no | Pain: VAS Disability: - |
| 25 | Park Sam-Ho 2022 [48] | RCT | N = 14 NCLBP+ instability Age: 31.07 ± 6.82 Male: 9 Female: 5 | • 5 weeks • 15 treatment sessions • Stabilization exs & respiratory resistance exs | N = 15 NCLBP+ instability Age: 30.29 ± 5.34 Male: 6 Female: 9 | • 5 weeks • 15 sessions • Stabilization exs | no | no | Pain: VAS Disability: RMQ |
| 26 | Masroor 2023 [49] | Sing-center Single-blind RCT | N = 11 CLBP Age: 25.3 ± 3.4 Male: N/A Female: N/A | • 4 weeks • 12 treatment sessions • Stabilization exs & Diaphragm breathing | N = 11 CLBP Age: 25.6 ± 4.2 Male: N/A Female: N/A | • 4 weeks • 12 treatment sessions • Stabilization exs | yes | no | Pain: NRS Disability: ODI |
| 27 | Siglan 2023 [50] | Single-center Double blind RCT | N = 21 CLBP Age: 38.76 ± 8.96 Male: 8 Female: 13 | • 4 weeks • 12 sessions • Physical therapy & release | N = 21 CLBP Age: 38 ± 7.78 Male: 12 Female: 9 | • 4 weeks • 12 sessions • Physical therapy & sham release | yes | yes | Pain: NRS Disability: RMQ |
| 28 | Salah Eldeen 2024 [51] | Sing-center RCT | N = 17 LBP Age: 37.81 ± 8.49 Male: 5 Female: 12 | • 6 weeks • 18 sessions • Physical therapy & Thoracic mobilization (T5-T12) | N = 17 LBP Age: 34.94 ± 4.71 Male: 9 Female: 8 | • 6 weeks • 18 sessions • Physical therapy | no | no | Pain: VAS Disability: - |

*(Continued)*

**Table 1.** (Continued)

| | Author-year | Study design | Treatment group characteristics | Treatment duration & Type | Control group characteristics | Control duration & Type | Blinding | | Outcome measure |
|---|---|---|---|---|---|---|---|---|---|
| | | | | | | | Patient | Assessor | |
| 29 | Dogan 2024 [52] | Single-center Double blind RCT | N = 27 CLBP Age: 36.76 ± 4.32 Male: 12 Female: 15 | • 6 weeks • 18 sessions • Stabilization exs & Thoracic mobility exs | N = 27 CLBP Age: 35.56 ± 7.55 Male: 14 Female: 13 | • 6 weeks • 18 sessions • Stabilization exs | no | yes | Pain: VAS Disability: QBPDS |
| 30 | Elfayoumi 2024 [53] | Single-center Single blind RCT | N = 21 CLBP Age: 25.19 ± 2.87 Male: 9 Female: 12 | • 4 weeks • 12 sessions • Stretching and strengthening exs & thoracic lymphatic pump technique | N = 21 CLBP Age: 25.43 ± 2.84 Male: 12 Female: 9 | • 4 weeks • 12 sessions • Stretching and strengthening exs | no | no | Pain: NRS Disability: ODI |
| 31 | Hwang 2024 [54] | Sing-center RCT | N = 16 NCLBP Age: 56.88 ± 6.93 Male: N/A Female: N/A | • 4 weeks • 12 sessions • Thoracic mobility exs | N = 16 NCLBP Age: 55.88 ± 6.61) Male: N/A Female: N/A | • 4 weeks • 12 sessions • Physical therapy | no | no | Pain: NRS Disability: ODI |
| 32 | Mikkonen 2024 [55] | Single blind RCT | N = 9 NCLBP Age: 50.1 ± 11.9 Male: 3 Female: 6 | • 8 weeks • Every day • MVT control exs & synchronized breathing | N = 13 NCLBP Age: 53.5 ± 9.7 Male: 5 Female: 8 | • 8 weeks • Every day • MVT control exs | yes | yes | Pain: NRS Disability: RMQ |
| 33 | Park Dongh-wan 2024 [56] | Single-center Single blind RCT | N = 15 NCLBP Age: 44.2 ± 5.09 Male: 8 Female: 7 | • 6 weeks • 18 sessions • Physical therapy & Thoracic mobility exs | N = 15 NCLBP Age: 42.6 ± 7.41 Male: 7 Female: 8 | • 6 weeks • 18 sessions • Physical therapy & Stabilization exs | no | yes | Pain: VAS Disability: - |
| 34 | Abdel-Aziz 2025 [57] | Single-center Single blind RCT | N = 27 CLBP Age: 28.21 ± 4.12 Male: - Female: 27 | • 6 weeks • 18 sessions • Physical therapy & Diaphragm release and 5 deep breaths | N = 27 CLBP Age: 29.93 ± 3.16 Male: - Female: 27 | • 6 weeks • 18 sessions • Physical therapy | yes | no | Pain: - Disability: ODI |
| 35 | Abdelha-mid 2025 [58] | Single-center Single blind RCT | N = 30 NCLBP Age: 21.43 ± 2.16 Male: 16 Female: 14 | • 4 weeks • 12 sessions • Stabilization exs & Ball and balloon exs | N = 30 NCLBP Age: 21.30 ± 2.45 Male: 15 Female:15 | • 4 weeks • 12 sessions • Stabilization exs | no | yes | Pain: VAS Disability: ODI |
| 36 | Li 2025 [59] | RCT | N = 6 NCLBP Age: 23.83 ± 2.48 Male: 2 Female: 4 | • 12 weeks • Home exs • Stabilization exs & Breathing exs | N = 6 NCLBP Age: 25.14 ± 1.35 Male: 2 Female: 4 | • 12 weeks • Home exs • Stabilization exs | no | no | Pain: VAS Disability: ODI |

RCT: Randomized Control Trial, NCLBP: Non-Specific Low Back Pain, VAS: Visual Analogue Scale, NPRS: Numerical Pain Rate Scale, ODI: Oswestry Disability Index, RMQ: Roland Morris Questionnaire, QBPDS: Quebec Back Pain Disability Scale, MVT: Movement.

## Assessment of publication bias

The assessment of reporting bias was performed using the Egger's weighted regression test [68] and Egger's publication bias graph. In addition, the Duval and Tweedie "trim and fill" approach was used to examine the potential influence of publication bias [70].

## Sensitivity analysis

The Jackknife (leave-one-out) approach was used to evaluate the impact of each individual research on the combined results [71]. In addition, a sensitivity analysis was conducted through performing a subgroup analysis on high-quality and higher-designed studies inside the meta-analysis. This was done to evaluate any possible alterations in the results.

## Data synthesis

All trials included in the analysis were considered appropriate for the meta-analysis because of their clinical homogeneity, precisely because they were randomized trials with a control group. The meta-analysis of study outcomes was performed using Stata software (version 14; Stata Corp., College Station, TX, USA).

The quality of evidence for study outcomes was assessed using the Grading of Recommendations, Assessment, Development, and Evaluation [72] and classified as high, moderate, low, and very low [73].

# Results

## Studies characteristics

A total of 18307 references were retrieved from the study selection process, as shown in Fig 1. Due to duplication, 9081 references were excluded. A total of 9226 articles were assessed for potential eligibility based on of their title and abstracts. After a thorough title and abstract review, 300 papers had full-text retrieval to evaluate their eligibility criteria. Subsequently, 42 research papers (five in Korean [29,37,47,54,56], one in Chinese [35] and one in Spanish [74]) were selected to be included in the implementation of this systematic review. In addition, six studies were excluded due to incomplete data on their outcome measures [74–79], leaving a total of 36 studies to be included in the analysis. The characteristics of the included studies are summarized in detail in Table 1. The studies were published between 2005 and 2025 with a total sample size of 1833 patients. All non-English studies (Korean, Chinese, Spanish) were accurately translated, ensuring the integrity and reliability of the extracted data.

Among the 36 studies included in the review, 31 [4,26–28,32–56,58,59] reported pain outcomes and 24 [13,26,29–31,33,35,36,38,39,41–43,46,48–50,52–55,57–59] reported disability outcomes, with some studies reporting only pain or only disability. Eleven studies [26,29,35,37,40,44,49,55,58,59,30] used breathing therapy as a treatment for CLBP. Four studies [27,36,39,47] evaluated the efficacy of thoracic manipulation, while six studies [28,31,34,41,42,51] implemented a treatment based on thoracic mobilization. Respiratory muscle training [32,33,38,4,43,45,46,48] was used as an intervention in eight studies. One study [13] used breathing exercises and thoracic mobilization simultaneously in its research, while another two [50,57] employed release methods in conjunction with standard physiotherapy. Three studies used thoracic mobility exercises as their intervention [52,54,56], while one study [53] benefited from thoracic lymphatic pump techniques, which involved a combination of manual and breathing techniques on the rib cage.

## Risk of bias assessment

The Cochrane Risk of Bias Questionnaire 2 (RoB 2) was used to assess the quality of 36 selected studies. Two studies [28,32] (5.6%) had high risk of bias, 22 studies [4,13,29–31,34,35,37,38,41–48,51,55,56,58,59] (61.1%) had some level of concern, and 12 studies [26,27,33,36,39,40,49,50,52–54,57] (33.3%) had low risk of bias. The level of inter-rater agreement for quality assessment was substantial or good according to the overall Gwet's score (KG: 0.66±0.07). Details of the risk of bias assessment for clinical trials are shown in Fig 2. Unclear randomization and deviations from the intended interventions were the main issues of high risk of bias or some concerns.

## Study design and outcome measurement

Table 1 details characteristics of studies included in this review regarding study design, treatment group characteristics, treatment duration and type, control group characteristics, control duration and type, assessor or patient blinding, and outcome measures.

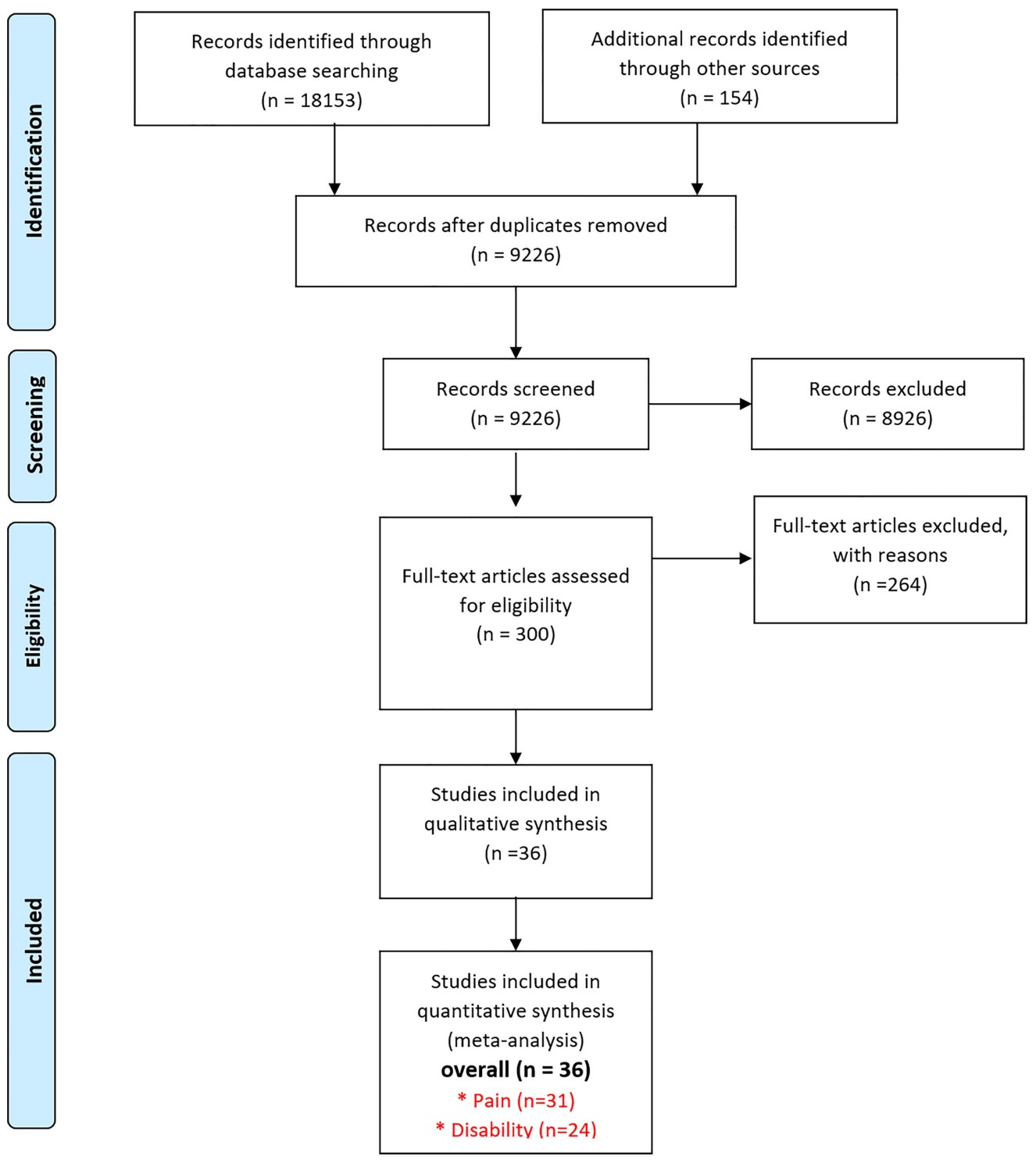

**Fig 1. Flow diagram for the study selection process.**

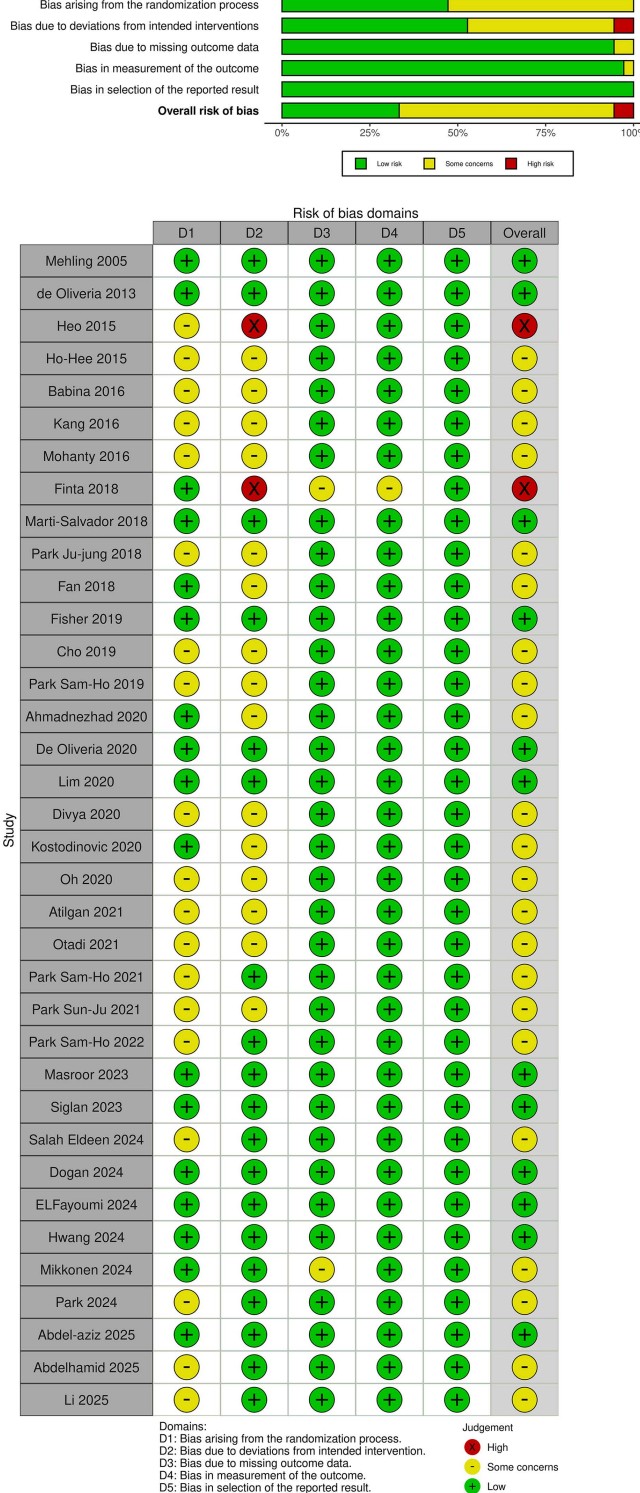

Fig 2. Summary of the risk of bias assessment: the review authors' judgments about each item of risk of bias for each included study.

### Effects of interventions on pain intensity

Among the included studies, 31 studies [4,26–28,32–56,58,59] reported the effect of interventions on patients' pain levels as one of their outcome measures. According to the data derived from the analysis, the pooled Morris' dppc for pain showed the statistically significant with small effect in reducing pain in the group of patients with breathing exercises and/ or thoracic techniques compared to the control group (dppc = −0.35, 95% CI = −0.46 to −0.23) with a considerable heterogeneity among these studies ($\chi^2$ = 200.29, $I^2$ = 85.0%, p = 0.00) (Fig 3). To identify the source(s) of heterogeneity, subgroup analysis was performed on eight potential factors (details in Table 2).

### Subgroup analysis on the effects of interventions on pain intensity

The included studies used different treatment methods. Thirteen studies used only breathing therapy (BT) as the primary treatment, while ten studies used thoracic techniques alone or in combination with breathing therapy (TT). The results of the subgroup analysis indicated a substantial moderate effect of TT in reducing pain (dppc = −0.42, 95% CI = −0.57 to −0.27) compared to BT (dppc = −0.21, 95% CI = −0.41 to −0.01). However, the overall heterogeneity remains constant within each subgroup (Table 2). Treatment more prolonged than four weeks and treatments more than twelve sessions showed a moderate effect in reducing pain intensity (dppc = −0.60, 95% CI = −0.81 to −0.39) (dppc = −0.63, 95% CI = −0.83to −0.43); however, they did not affect the amount of heterogeneity (Table 2).

A large effect size was found in a subgroup of patients who were unaware of the type of therapy they received (dppc = −0.72, 95% CI = −0.94 to −0.49). Additionally, trials with assessor blinding had moderate to large effects on pain intensity (dppc = −0.64, 95% CI = −0.82 to −0.46). The amount of heterogeneity did not change meaningfully in these two post hoc analyses. Regarding sample size, studies with fewer than 20 participants were associated with a larger effect size (dppc = −0.69, 95% CI = −0.91 to −0.46), and the level of heterogeneity decreased by 15.2% in the subgroup of studies with smaller sample sizes ($\chi^2$ = 49.71, $I^2$ = 69.8%, p = 0.00). Subgroup analysis by patients' age (younger or older than 40 years old) did not change the effect size or heterogeneity.

According to the risk of bias assessment, most of the included studies had some concerns, and this subgroup showed a greater effect size (dppc = −0.58, 95% CI = −0.77 to −0.39) compared to studies with a high or low risk of bias. Additionally, the subgroup of studies with a low risk of bias showed somewhat less heterogeneity ($\chi^2$ = 19.05, $I^2$ = 73.2%, p = 0.00).

### Effects of interventions on disability score

Out of the studies included in this meta-analysis, twenty-four studies [13,26,29–31,33,35,36,38,39,41–43,46,48–50,52–55,57–59] evaluated the level of disability as their outcome measure. The pooled Morris' dppc calculated for disability indicated a statistically significant improvement with large effect compared to the control group (dppc = −0.71, 95% CI = −0.86 to −0.57). Nevertheless, there is a significant degree of heterogeneity among the studies ($\chi^2$ = 266.16, $I^2$ = 91.4%, p = 0.00) (Fig 4). Table 3 presents the results of the subgroup analysis of potential covariates.

### Subgroup analysis on the effects of interventions on disability score

Twelve trials used BT as the primary treatment, while twelve other trials used TT. A subgroup analysis revealed that combining data from trials employing the TT approach yielded approximately twice the improvement in disability scores (dppc = −0.84, 95% CI = −1.02 to −0.67) compared to trials using only BT (dppc = −0.39, 95% CI = −0.67 to −0.12). Subgroup analysis by treatment weeks showed a larger effect for treatments lasting more than four weeks (dppc = −0.85, 95% CI = −1.11 to −0.59), with a significant reduction in heterogeneity to an ignorable amount ($\chi^2$ = 11.50, $I^2$ = 30.5%, p = 0.00). However, studies with more than 12 treatment sessions had a larger effect size (dppc = −1.05, 95% CI = −1.31 to −0.78), but this had no impact on heterogeneity ($\chi^2$ = 84, $I^2$ = 93.9%, p = 0.00) (Table 3).

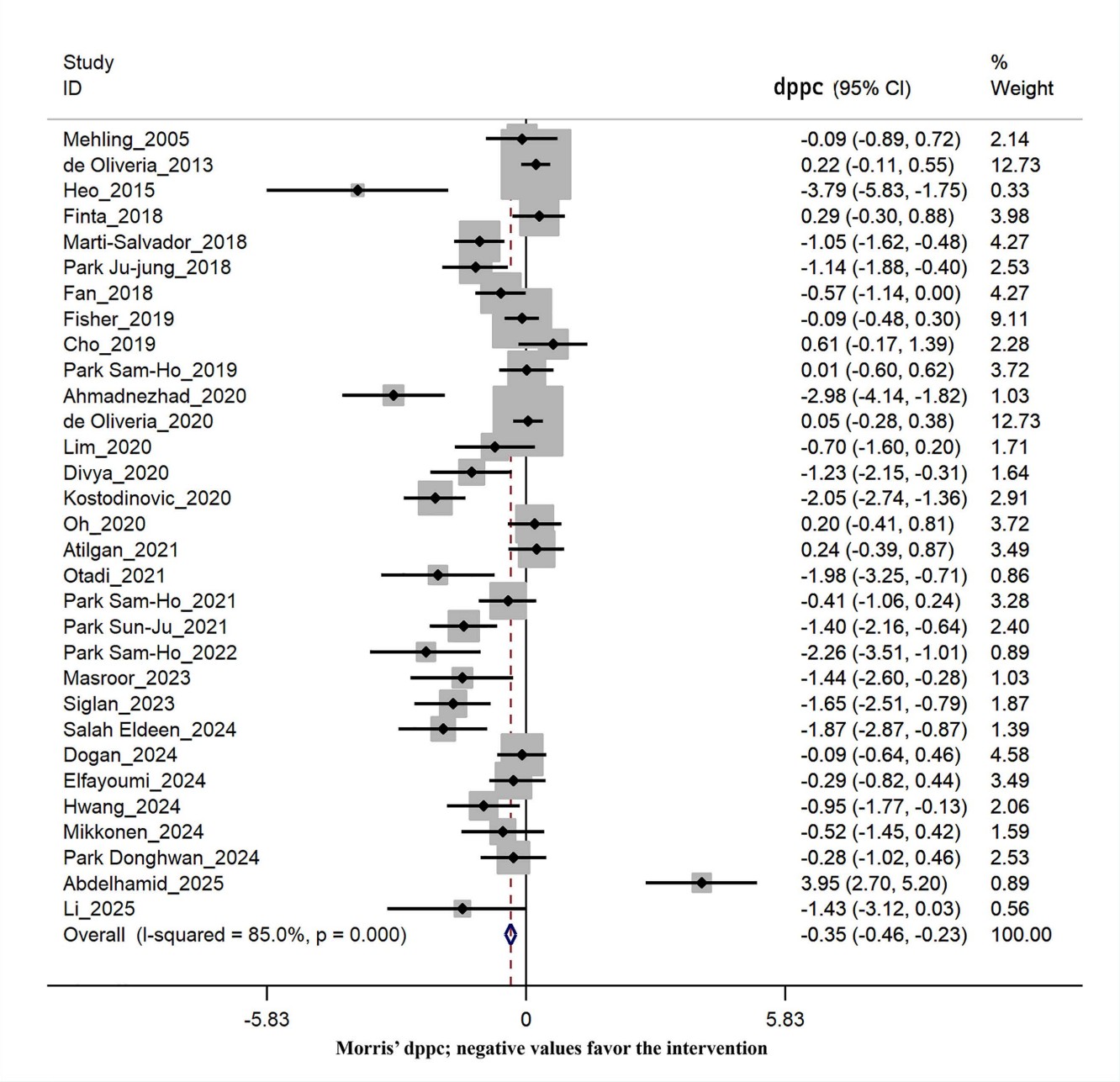

**Fig 3. Forest plot of the effect of the breathing exercises and/or thoracic techniques group compared to the control group on the intensity of pain (Morris' dppc; negative values favor the intervention).**

Trials with blinded patients showed a greater effect on the disability score (DPC = −0.85, 95% CI = −1.07 to −0.64), with a meaningful decrease in heterogeneity to a non-significant level ($\chi^2$ = 11.53, $I^2$ = 39.3%, p = 0.00). However, when considering subgroups of studies with assessor blinding, the effect size diminished and the heterogeneity remained constant (dppc = −0.47, 95% CI = −0.66 to −0.28). Studies with more than 20 samples, however, showed a smaller effect size

**Table 2. Subgroup analysis for the effect of potential factors on pain intensity.**

| Potential | | Morris' dppc (95%CI) | NO of studies | Heterogeneity χ² | Pvalue | I² |
|---|---|---|---|---|---|---|
| Main treatment | Just Breath therapy | −0.21 (−0.41 to −0.01) | 16 | 103.61 | 0.00 | 85.5% |
| | Thoracic techniques with/without breath therapy | −0.42 (−0.57 to −0.27) | 15 | 93.83 | 0.00 | 85.1% |
| Treatment weeks | >4 weeks | −0.60 (−0.81 to −0.39) | 15 | 91.28 | 0.00 | 84.7% |
| | =<4weeks | −0.23 (−0.37 to −0.09) | 16 | 100.77 | 0.00 | 85.1% |
| Treatment sessions | >12se | −0.63 (−0.83 to −0.43) | 15 | 89.68 | 0.00 | 84.4% |
| | =<12se | −0.20 (−0.35 to −0.06) | 16 | 99.36 | 0.00 | 84.9% |
| Patient blinding | yes | −0.72 (−0.94 to −0.49) | 8 | 38.01 | 0.00 | 81.6% |
| | no | −0.20 (−0.34 to −0.06) | 23 | 147.54 | 0.00 | 85.1% |
| Assessor blinding | yes | −0.64 (−0.82 to −0.46) | 12 | 95.13 | 0.00 | 88.4% |
| | no | −0.12 (−0.28 to 0.03) | 19 | 87.04 | 0.00 | 79.3% |
| Sample size | >20 | −0.21 (−0.35 to −0.07) | 15 | 137.80 | 0.00 | 89.8% |
| | =<20 | −0.69 (−0.91 to −0.46) | 16 | 49.71 | 0.00 | 69.8% |
| Age | >40 yrs. | −0.31 (−0.46 to −0.16) | 15 | 76.93 | 0.00 | 81.8% |
| | <40 yrs. | −0.40 (−0.58 to −0.21) | 16 | 122.81 | 0.00 | 87.8% |
| Risk of bias | High | −0.02 (−0.59 to 0.54) | 2 | 58.54 | 0.00 | 92.9% |
| | Some concern | −0.58 (−0.77 to −0.39) | 18 | 85.77 | 0.00 | 87.7% |
| | low | −0.20 (−0.36 to −0.04) | 11 | 19.05 | 0.00 | 73.2% |
| All studies | | −0.35 (−0.46 to −0.23) | 31 | 200.29 | 0.00 | 85.0% |

(dppc = −0.66, 95% CI = −0.83 to −0.50) than studies with fewer than 20 samples. Post hoc analysis of patient age did not alter the overall findings regarding effect size or heterogeneity (Table 3).

None of the studies included in the meta-analysis of treatment effects on disability scores were considered to be at high risk for bias. Studies with some concern of bias had a larger effect size (dppc = −0.96, 95% CI = −1.21 to −0.71, p = 0.00) than studies with a low risk of bias (dppc = −0.58, 95% CI = −0.76 to −0.41, p = 0.00). The level of heterogeneity was smaller in the subgroup of studies with a low risk of bias (χ² = 38.78, I² = 71.0%, p = 0.00; see Table 3).

## Sensitivity analysis

**Pain Intensity:** The presence of bias in the research affects the assessment of the treatment effects on patients' pain severity. The analysis of studies with a low risk of bias regarding pain revealed a slight decrease in effect size. The same was found for the pooled Morris' dppc for the trials with not small sample size (Table 4).

**Disability Score:** A sensitivity analysis of low-risk-of-bias and blinded trials revealed a reduction in the magnitude of the treatment effect on patient disability scores, from large to moderate. (Table 4).

**Jackknife Method:** The effect of 1 study on the estimate of the combined meta-analysis was examined using the leave-one-out method of sensitivity analysis. None of the studies affected the overall combination of the pain intensity or disability score outcomes, except for one study. After removing the Ho-Hee [29] study, the estimated pooled Morris' dppcs of disability scores changed to −0.58 (95% CI: −0.68, −0.49) (Fig 5).

## Assessment of publication bias

Egger's linear regression did not confirm the presence of substantial publication bias for either pain intensity or disability scores (Fig 6). The application of the trim-and-fill technique also failed to detect any missing studies. However, funnel plots indicated a significant presence of publication bias in the pain intensity studies and disability scores (Fig 7).

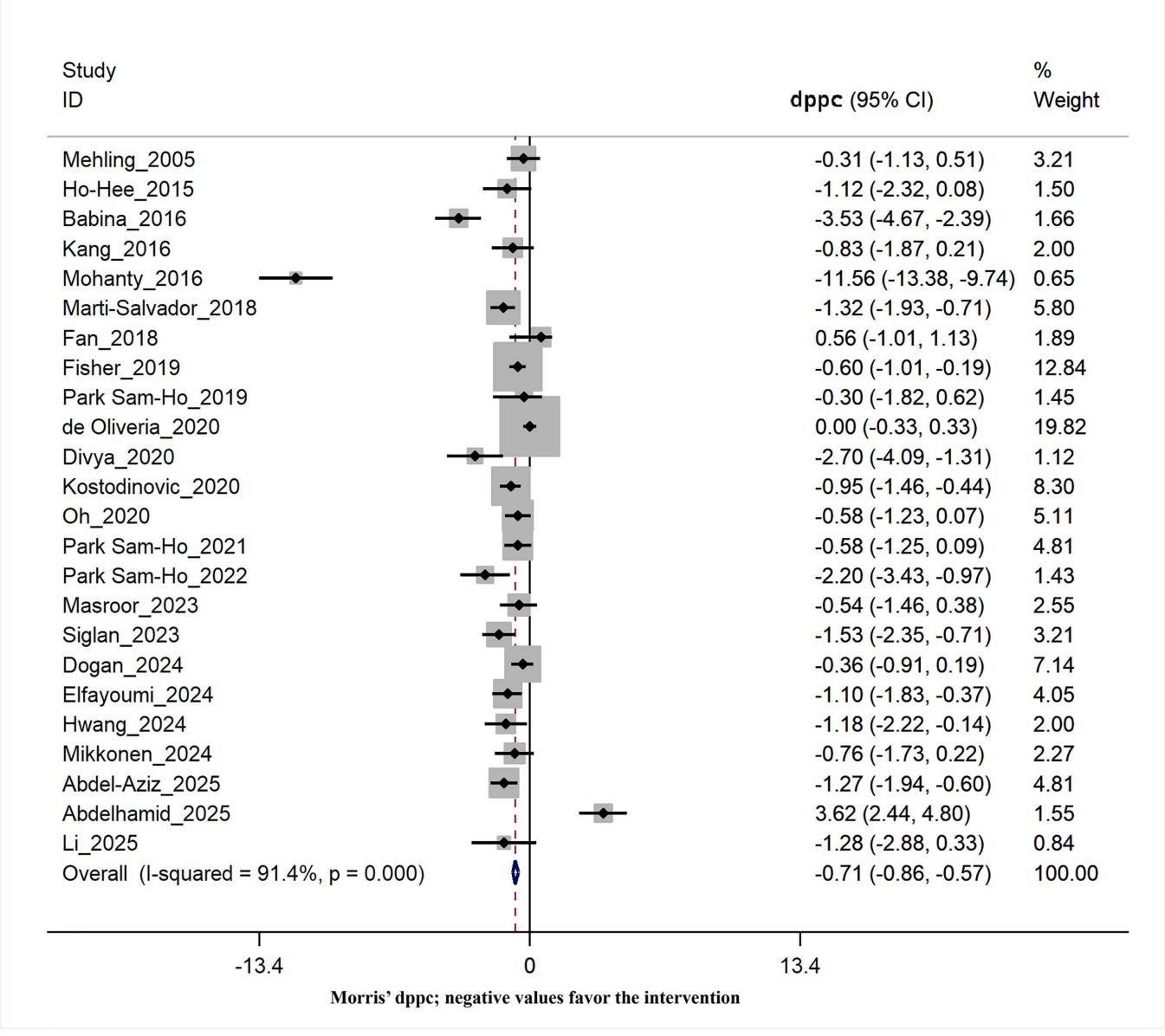

**Fig 4. Forest plot of the effect of the breathing exercises and/or thoracic techniques group compared to the control group on the disability score (Morris' dppc; negative values favor the intervention).**

## Quality of evidence

We evaluated the quality of the evidence for the outcome measures using the GRADE methodology (Table 5). Regarding the effect of breathing exercises and/or thoracic techniques on pain in patients with low back pain, the quality of the evidence was very low (downgraded due to limitations, inconsistencies, and publication bias, and upgraded due to the large effect size). For the outcome disability, the quality of the evidence was also low (downgraded due to limitations and inconsistency, and upgraded due to the large effect size).

**Table 3. Subgroup analysis for the effect of potential factors on disability score.**

| Potential | | Morris' dppc (95%CI) | NO of studies | Heterogenity χ2 | Pvalue | I² |
|---|---|---|---|---|---|---|
| Main treatment | Just Breath therapy | −0.39 (−0.67 to −0.12) | 12 | 60.35 | 0.00 | 81.8% |
| | Thoracic techniques with/without breath therapy | −0.84 (−1.02 to −0.67) | 12 | 198.43 | 0.00 | 94.5% |
| Treatment weeks | >4 weeks | −0.85 (−1.11 to −0.59) | 9 | 11.50 | 0.00 | 30.5% |
| | =<4weeks | −0.65 (−0.83 to −0.47) | 15 | 253.06 | 0.00 | 94.5% |
| Treatment sessions | >12se | −1.05 (−1.31 to −0.78) | 10 | 84.00 | 0.00 | 93.9% |
| | =<12se | −0.56 (−0.74 to −0.39) | 14 | 144.48 | 0.00 | 88.2% |
| Patient blinding | yes | −0.85 (−1.07 to −0.64) | 16 | 11.53 | 0.00 | 39.3% |
| | no | −0.59 (−0.79 to −0.39) | 8 | 251.60 | 0.00 | 94.0% |
| Assessor blinding | yes | −0.47 (−0.66 to −0.28) | 9 | 22.91 | 0.00 | 91.7% |
| | no | −1.07 (−1.30 to −0.84) | 15 | 237.33 | 0.00 | 90.9% |
| Sample size | >20 | −0.66 (−0.83 to −0.50) | 11 | 248.96 | 0.00 | 95.2% |
| | =<20 | −0.87 (−1.18 to −0.57) | 13 | 15.81 | 0.00 | 36.7% |
| Age | >40 yrs. | −0.70 (−0.90 to −0.51) | 12 | 189.97 | 0.00 | 94.2% |
| | <40 yrs. | −0.73 (−0.95 to −0.50) | 12 | 76.17 | 0.00 | 85.6% |
| Risk of bias | High | – | 0 | – | – | – |
| | Some concern | −0.96 (−1.21 to −0.71) | 14 | 219.73 | 0.00 | 94.3% |
| | low | −0.58 (−0.76 to −0.40) | 10 | 38.78 | 0.00 | 71.0% |
| All studies | | −0.71 (−0.86 to −0.57) | 24 | 266.16 | 0.00 | 91.4% |

**Table 4. Sensitivity Analysis using any kinds of blinding and studies with low risk of bias.**

| Pain Intensity | | | | |
|---|---|---|---|---|
| Combination | Number | Morris' dppc | Lower limit | Upper limit |
| Low risk of bias | 11 | −0.20 | −0.36 | −0.04 |
| Sample size (>20) | 15 | −0.21 | −0.35 | −0.07 |
| All study | 31 | −0.34 | −0.46 | −0.23 |
| **Disability Score** | | | | |
| Low risk of bias | 10 | −0.58 | −0.76 | −0.40 |
| Sample size (>20) | 11 | −0.66 | −0.83 | −0.50 |
| All study | 24 | −0.71 | −0.86 | −0.57 |

RCT: Randomized Control Trial.

## Discussion

This systematic review and meta-analysis examined the effects of breathing exercises and thoracic techniques on pain and disability in patients with low back pain. Our study is the first to evaluate the impact of thoracic-focused interventions on LBP outcomes, highlighting the clinical importance of considering the thoracic region in rehabilitation strategies. Overall, these interventions modestly reduced pain but demonstrated a larger effect on disability. However, the interpretation of these findings must remain cautious given the overall low certainty of evidence ant the observed heterogeneity across included studies. Although subgroup analyses for treatment duration and methodological quality reduced heterogeneity for disability outcomes, significant variability remain for pain outcomes. This suggests that unmeasured moderators may influence the observed effects.

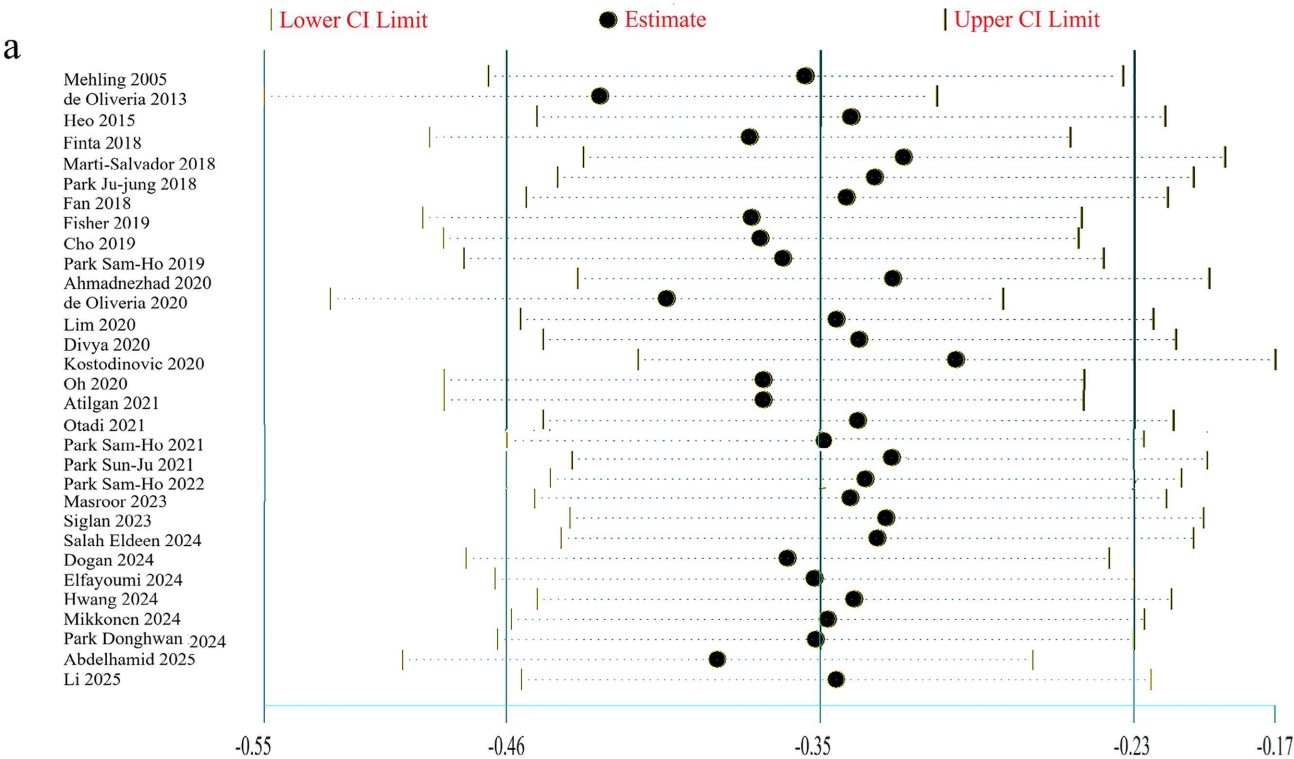

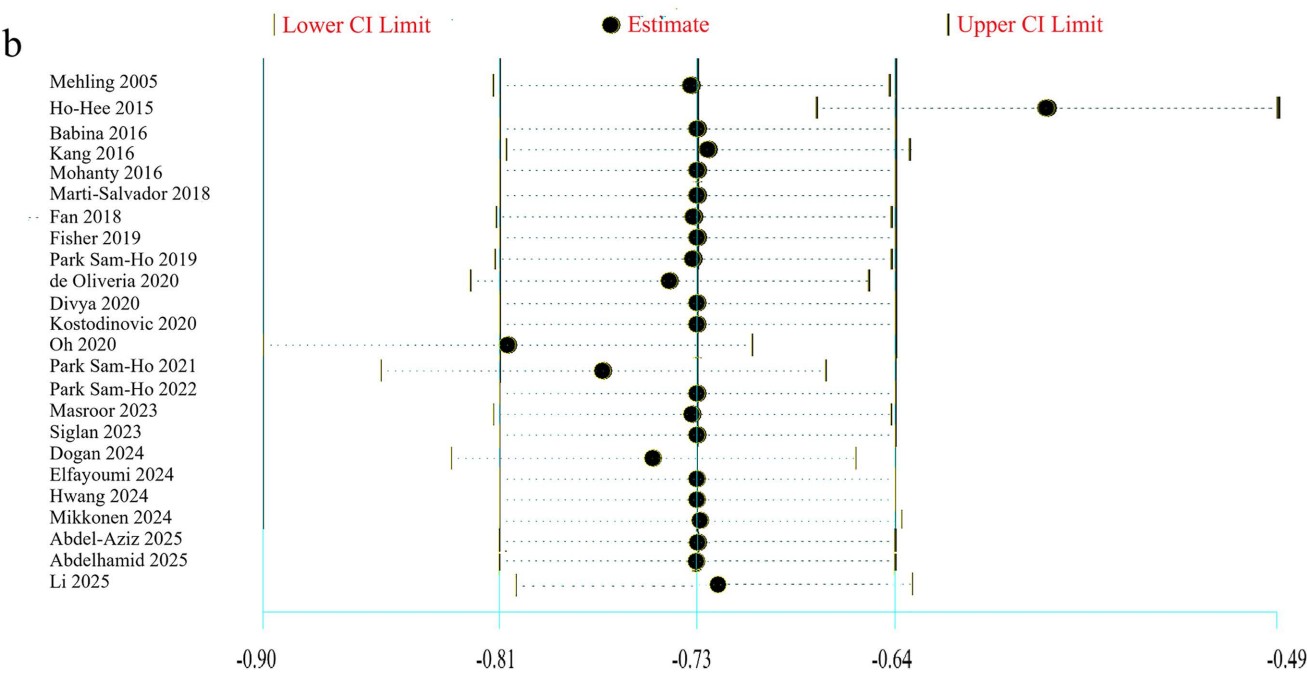

**Fig 5. Influence analyses of each separate study on the overall estimates of a) Pain intensity b) Disability score.**

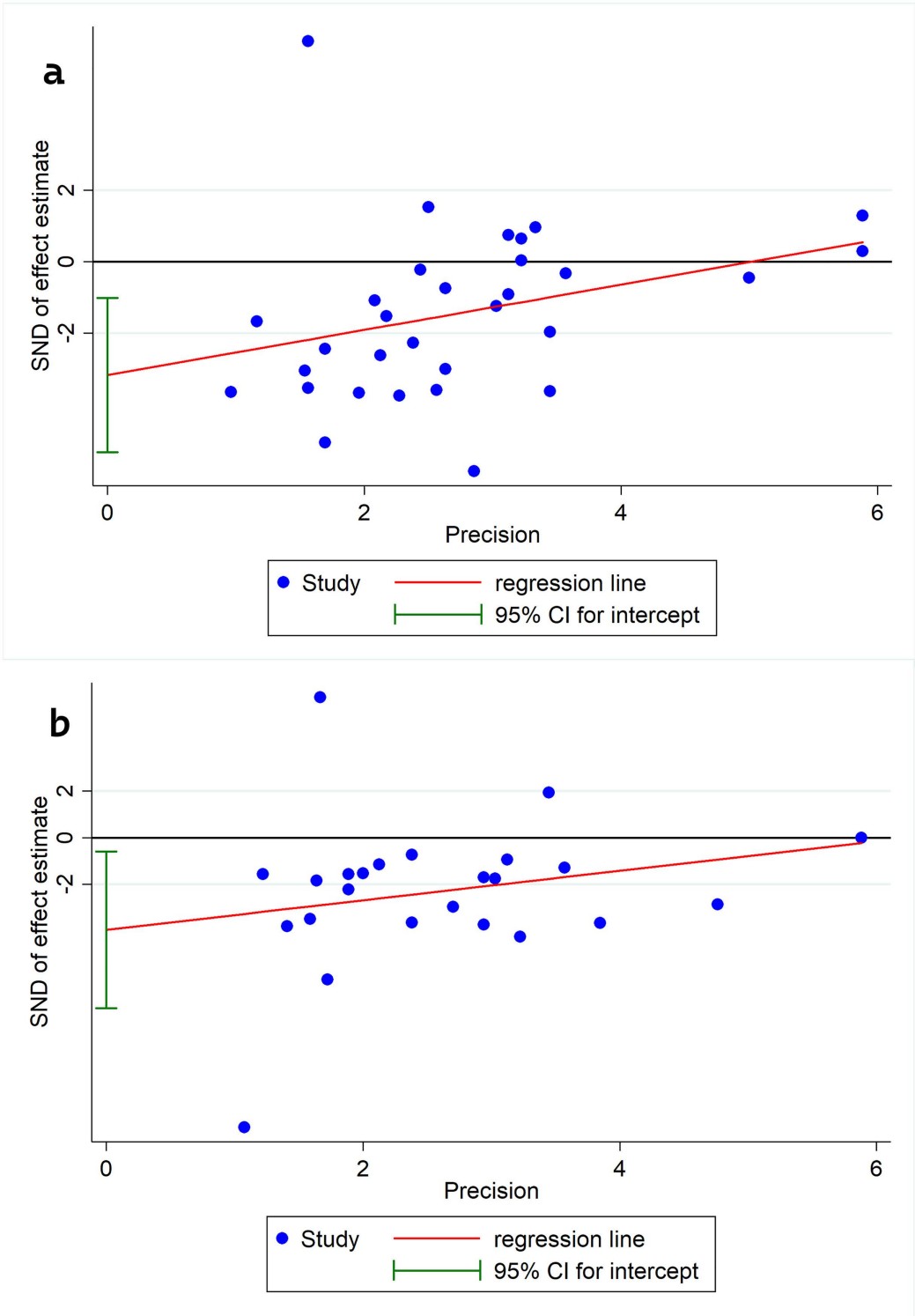

**Fig 6. Egger's graphs for determining publication bias in the meta-analysis for: a) Pain intensity b) disability score.**

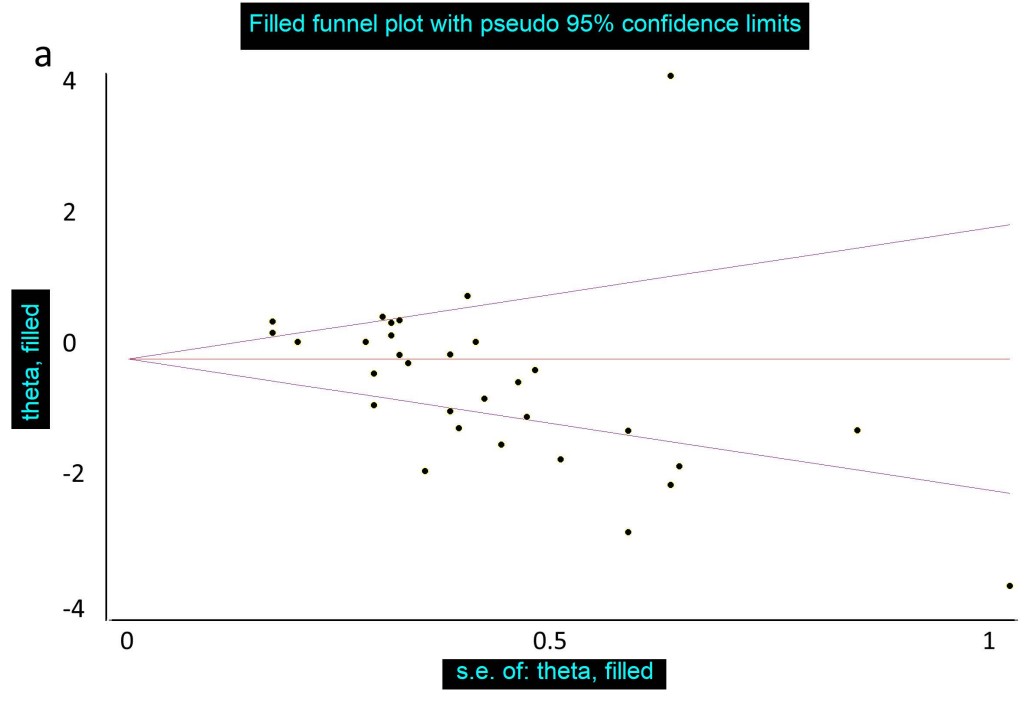

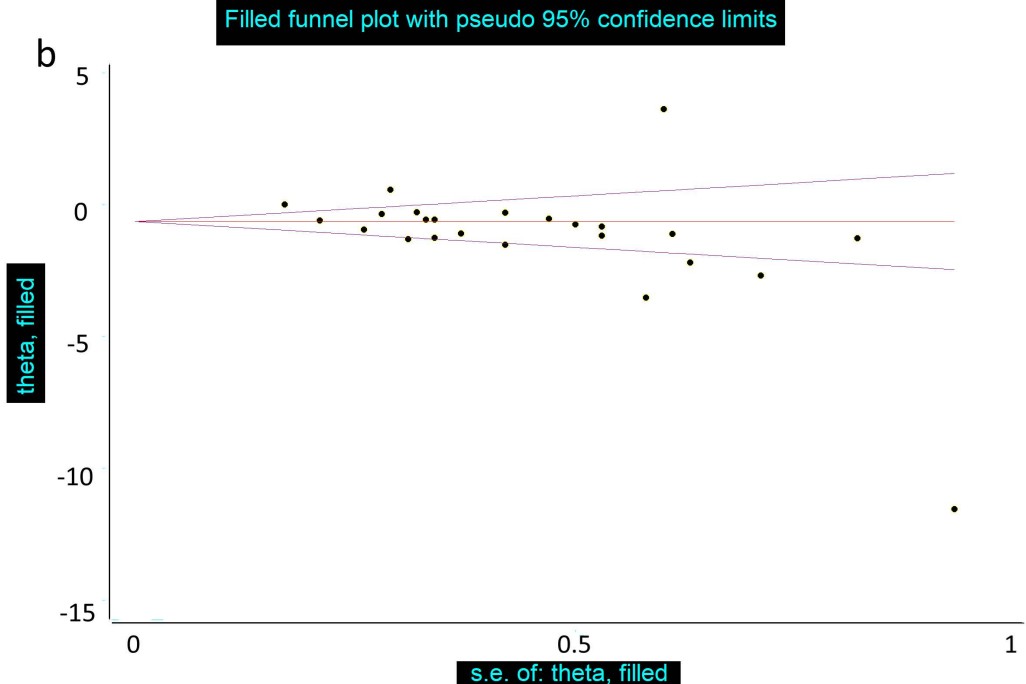

**Fig 7. Funnel plot for determining publication bias in the meta-analysis for: a) Pain intensity b) disability score.**

**Table 5. Grades of Recommendation, Assessment, Development and Evaluation quality of evidence for the primary outcomes.**

| Outcome | Limitations | Inconsistency | Indirectness | Imprecision | Publication Bias | Effect size | GRADE Quality |
|---|---|---|---|---|---|---|---|
| Pain | −1* | −2† | 0 | 0 | −1‡ | 0 | low |
| Disability | −1* | −1† | 0 | 0 | −2‡ | +2£ | Very low |

* Downgraded by one level as the majority of the studies have a high risk of bias and some concerns.

† Two levels were downgraded. The I² value was > 50% for pain. One level was downgraded. The I² value was > 50% but I2 was < 50% in some sub-groups.

‡ Downgraded by due to suspected publication bias.

£ Upgraded two level due to large effect size.

Our findings align with and extend the growing body of recent evidence supporting the role of respiratory-focused interventions in low back pain management. Shi et al. (2023) [16] and Jiang et al. (2024) [15] both demonstrated that breathing exercises lead to statistically significant but modest improvements in pain and disability, consistent with the small-to-moderate pooled effects observed in our study. Similarly, Zhai et al. (2024) [17] highlighted those respiratory interventions are more effective when delivered beyond four weeks, a result that resonates with our subgroup analysis showing that longer treatment duration reduces heterogeneity and yields more consistent improvements in disability. Importantly, Fabero-Garrido et al. (2024) [18] reported that respiratory muscle training not only improves pain and disability but also enhances respiratory function and functional ability, underscoring the broader physiological and biomechanical relevance of respiratory-targeted rehabilitation strategies. Additionally, systematic reviews of general manual therapy also reported positive effects [20,80]. Collectively, these recent reviews reinforce our conclusion that breathing exercises and thoracic manual techniques may provide meaningful adjunctive benefits in the management of low back pain, though the certainty of these effects remains limited and should be interpreted with caution.

The diaphragm plays a crucial role in spinal stability and interacts synergistically with the transverse abdominis. Altered activation patterns in these muscles have been reported in individuals with CLBP [81]. Dysfunction of the diaphragm as the primary respiratory muscle can result in disordered breathing patterns, hyperventilation, and hypocapnia, negatively affecting muscular function [82]. Approximately 50–60% of individuals with CLBP exhibit altered breathing during trunk stability testing, linked to impaired motor control rather than pain intensity [83].The clinical implications of our findings suggest that integrating thoracic mobilization with breathing exercises may provide superior outcomes.

Several subgroups were used to identify and account for influential elements of the research results, including treatment type, treatment duration, patient blinding, rater blinding, sample size, average age of participants, and risk of bias in the studies. Within a subgroup categorized by treatment technique, studies using breath therapy showed a small effect on reducing pain and disability. In contrast, studies that used a thoracic technique approach showed much larger effect sizes, with moderate and large sizes observed for pain and disability scores, respectively. These results indicate that the thoracic techniques produced superior improvements in both pain and disability in people with LBP, emphasizing their clinical usefulness compared with breathing therapy alone. However, the assessment of the degree of heterogeneity did not change within the subgroups. Therefore, these findings should be viewed as exploratory and hypothesis-generating rather than confirmatory, given the residual variability and low certainty of evidence.The more significant treatment effects of the thoracic technique may be related to increased thoracic spine mobility, which is more accessible with this technique. Restricted motion of nearby joints may cause lumbar motion dysfunction and resultant pain. Restricted hip rotation has previously been reported in LBP [84]. Similarly, excessive lumbar rotation and back pain may result from limited thoracic spine motion [85]. These findings support the concept that restoring thoracic mobility can reduce compensatory lumbar overload and improve overall spinal function. Babina et al. [13] observed that participants who underwent thoracic mobilization together with specific therapy for LBP demonstrated considerably greater improvement in chest wall expansion. This improvement can be attributed to the mechanical effects of

posterior-anterior mobilization applied to the thoracic region. Furthermore, breathing is a biological activity involving the muscles and skeleton that is essential to the functioning of our internal organs. Breathing plays a vital role in connecting the body's physical and internal systems [86]. In addition, manual therapy of the upper back regions may influence autonomic control of the lungs, which receive parasympathetic innervation from the vagus nerve and sympathetic innervation from the T1–T5 spinal segments. This mechanism could partly explain the observed improvement in respiratory function and breathing efficiency [87]. Manual techniques can effectively induce significant plastic deformation in connective tissue, thereby improving joint mobility [88]. Such movement-induced changes stimulate glycosaminoglycans, restore natural lubrication between collagen fibers and promote healthier tissue remodeling, which may contribute to increase contractile protein and oxidative capacity within muscle fibers [89]. While these mechanisms are plausible and supported by existing literature, we have tempered our interpretation to avoid overstating causal relationships. Future mechanistic studies are needed to confirm whether improvements in respiratory and thoracic function directly translate to clinical benefits in LBP.

Subgroup analysis in the current study suggests that increasing treatment duration leads to better outcomes. The timing and duration of treatment have been debated in previous studies. Zhai et al. [17] found that treatment beyond four weeks, delivered three to five times per week, had a greater impact on pain and disabilities reduction. However, Shi et al. [16] did not identify treatment length as a significant factor. In our analysis, restricting the studies to interventions longer than four weeks for disability outcomes reduced heterogeneity to approximately 30%, suggesting that treatment duration may partially explain variability between studies. This reduction indicates that longer treatment programs may yield more consistent and clinically meaningful improvements in disability. For pain outcomes, however, heterogeneity remained high ($I^2 > 85\%$), which may be due to additional unmeasured factors such as differences in chronicity of LBP, co-interventions, intensity of exercises, or timing of outcome assessments. These observations highlight the importance of standardizing treatment protocols and reporting practices in future studies. Collectively, these findings emphasize the need for future research to establish the most effective and efficient dosage for breathing therapy and thoracic techniques.

Blinding is another critical issue in clinical trials. Our results showed that patient blinding reduced the reported effect size for pain and decreased heterogeneity. While rater blinding, reduced treatment effects overall. Sample size also influenced findings: smaller studies tended to magnify effects on both pain and disability, and also reduced heterogeneity to below 50% for disability. In line with our findings, Divya et al. [41] noted that although the therapy group showed beneficial, the limited sample size may affect the validity of the final results. In contrast, patient age did not influence treatment effects. Our results confirmed the suggestion of previous studies [16] that study quality is an essential factor to mention; trials with lower methodological quality showed larger treatment effects, particularly for disability. This suggests that bias may overestimate the favorable treatment outcomes and the magnitude of the pooled effect size.

Our evaluation of publication bias yielded mixed findings. Egger's linear regression and the trim-and-fill method did not indicate substantial publication bias for either pain intensity or disability outcomes. However, funnel plot inspection suggested a possible presence of publication bias for both outcomes, reflecting potential small-study effects or heterogeneity in study protocols. Sensitivity analysis using the leave-one-out method showed that most individual studies did not significantly alter the overall pooled effect. The exception was the Ho-Hee [29] study, removal of which changed the pooled Morris' dppcs for disability to −0.58 (95% CI: −0.68 to −0.49). This indicates that this study had a notable influence on the combined estimate, likely due to its sample characteristics or methodology. These findings suggest that while the overall results are robust, caution is warranted in interpreting effect sizes, particularly for disability outcomes. Clinically, breathing exercises and thoracic techniques appear beneficial, but the magnitude of effect may be sensitive to individual studies. This underscores the need for future high-quality, adequately powered trials to confirm the effectiveness of these interventions.

## Strengths and limitations

This study included studies that targeted the thoracic region – through manipulation, mobilization, release, or breathing exercises. The study also performed subgroup analyses and included a broad time frame, more recent studies, and avoided language restrictions, all of which strengthen the comprehensiveness of this review.

However, heterogeneity across protocols—particularly in treatment duration, frequency, and exercise type—limited the meta-analysis. In addition, a meta-analysis based on total treatment time and also on the nature of the outcome measure (whether the outcomes were evaluated primarily or secondarily) might be more informative, but this was not possible due to incomplete reporting in the included study. Even after several subgroup analyses, heterogeneity for pain outcomes remained high, indicating unmeasured factors may have influenced the results. Specifically, differences in LBP chronicity may have contributed to variability, as acute, subacute, and chronic presentations can differ in underlying pathophysiology and recovery potential, affecting responsiveness to interventions. Co-interventions, such as concurrent physical therapy, exercise programs, or pharmacological treatments, may have confounded outcomes by producing additive or synergistic effects that varied across studies. Additionally, variability in the intensity, frequency, and progression of breathing exercises or thoracic techniques likely influenced treatment effects. These variables were not consistently reported in the included studies, and thus could not be analyzed in our review. Furthermore, comparison against minimal clinically important differences (MCID) were not feasible because studies used different scales. Instead, we calculated effect sizes using Morris's delta based on pre–post differences. Taken together, these limitations highlight the need for standardized intervention protocols, consistent outcome measures, and comprehensive reporting in future trials to reduce heterogeneity and improve interpretability.

## Conclusion

Based on the available evidence, breathing exercises and thoracic techniques appear more effective for reducing disability than pain in patients with LBP, with thoracic techniques demonstrating greater benefits overall. Interventions lasting more than four weeks may provide more consistent and clinically meaningful improvements in disability, highlighting the importance of adequate treatment duration. Nevertheless, these results should be interpreted cautiously due to the low certainty of evidence, residual heterogeneity, and limited exploration of moderating factors. Future research should focus on identifying key moderators (e.g., chronicity, co-interventions, exercise intensity) that may explain variability in outcomes, and on conducting high-quality, adequately powered randomized controlled trials using standardized protocols and consistent outcome measures. By addressing these methodological gaps, future studies can provide stronger evidence to guide the integration of thoracic and breathing interventions into physiotherapy practice for low back pain.

## Supporting information

**S1 Appendix. PRISMA checklist.**
(DOCX)

**S2 Appendix. Full search syntax.**
(DOCX)

**S1 File. Data excel file.**
(XLS)

## Acknowledgments

The authors would like to thank the Babol University of Medical Sciences for all the support they have provided.

## Author contributions

**Conceptualization:** Tahere Seyedhoseinpoor.

**Data curation:** Ramin Jafari, Zohreh Shafizadegan, Maryam Abbaszadeh-Amirdehi.

**Formal analysis:** Tahere Seyedhoseinpoor.

**Methodology:** Tahere Seyedhoseinpoor, Zohreh Shafizadegan.

**Supervision:** Tahere Seyedhoseinpoor.

**Writing – original draft:** Ramin Jafari.

**Writing – review & editing:** Tahere Seyedhoseinpoor, Zohreh Shafizadegan, Maryam Abbaszadeh-Amirdehi.

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
