## [Decision Letter · Decision Letter 0]

26 Sep 2025

Dear Dr. Seyedhoseinpoor,

Thank you for submitting your manuscript to PLOS ONE. After careful consideration, we feel that it has merit but does not fully meet PLOS ONE’s publication criteria as it currently stands. Therefore, we invite you to submit a revised version of the manuscript that addresses the points raised during the review process.

We look forward to receiving your revised manuscript.

Kind regards,

Seyed Hamed Mousavi

Academic Editor

PLOS ONE

**Journal Requirements:**

1. When submitting your revision, we need you to address these additional requirements. Please ensure that your manuscript meets PLOS ONE's style requirements, including those for file naming. The PLOS ONE style templates can be found at https://journals.plos.org/plosone/s/file?id=wjVg/PLOSOne_formatting_sample_main_body.pdf and https://journals.plos.org/plosone/s/file?id=ba62/PLOSOne_formatting_sample_title_authors_affiliations.pdf 2. Please include captions for your Supporting Information files at the end of your manuscript, and update any in-text citations to match accordingly. Please see our Supporting Information guidelines for more information: http://journals.plos.org/plosone/s/supporting-information. 3. If the reviewer comments include a recommendation to cite specific previously published works, please review and evaluate these publications to determine whether they are relevant and should be cited. There is no requirement to cite these works unless the editor has indicated otherwise. 

Reviewers' comments:

**Comments to the Author**

1. Is the manuscript technically sound, and do the data support the conclusions?

Reviewer #1: Yes

Reviewer #2: Yes

2. Has the statistical analysis been performed appropriately and rigorously?

Reviewer #1: Yes

Reviewer #2: Yes

3. Have the authors made all data underlying the findings in their manuscript fully available?

Reviewer #1: Yes

Reviewer #2: Yes

4. Is the manuscript presented in an intelligible fashion and written in standard English?

Reviewer #1: Yes

Reviewer #2: Yes

**Reviewer #1:**  Dear Authors

I would like to thank you for giving me an opportunity to consider this work for publication. You well done the a point by point answer to the comments of the reviewers about previous revision. Paper needs few minor edits actually now:

- I recommend adding a few sentences in the introduction, about the effectiveness and usefulness of the Effects of Breathing Exercise in various musculoskeletal disorders, neck pain, shoulder pain, low back pain.

- Discussions should be reviewed in light of the overall improvement of the paper. Redundant sentences and prewritten information should be avoided. Focus on take-home messages and how that information impacts the clinical practice of management these patients.

- I suggest to add this reference:

Cefalì A, Santini D, Lopez G, Maselli F, Rossettini G, Crestani M, Lullo G, Young I, Dunning J, de Abreu RM, Mourad F. Effects of Breathing Exercises on Neck Pain Management: A Systematic Review with Meta-Analysis. J Clin Med. 2025 Jan 22;14(3). doi: 10.3390/jcm14030709. Review. PubMed PMID: 39941380; PubMed Central PMCID: PMC11818914.

**Reviewer #2:**  This is a well-structured systematic review and meta-analysis examining the effects of breathing exercises and thoracic techniques on pain and disability in patients with low back pain (LBP). The study addresses an important clinical topic, given the high prevalence of LBP and the need for non-invasive interventions. The methods are rigorous, adhering to PRISMA guidelines, with a registered protocol on PROSPERO, comprehensive search strategy, appropriate risk of bias assessment using RoB 2, and quality evaluation via GRADE. The inclusion of subgroup and sensitivity analyses to address heterogeneity is commendable. However, the high heterogeneity in results, low quality of evidence, and potential publication bias limit the strength of conclusions. The revised version incorporates an updated search (to May 1, 2025) and additional studies, strengthening the analysis compared to prior iterations.

Major Comments:

1. Heterogeneity: High heterogeneity (I² > 85% for pain, >90% for disability) persists despite subgroup analyses. While some reductions occur (e.g., I² = 30.5% for >4 weeks treatment in disability), sources remain unclear. Discuss potential unmeasured factors (e.g., LBP chronicity, co-interventions, outcome measurement timing) more deeply. Consider meta-regression if data allow, or explicitly state why it was not performed. In the revised version, the shift to SMD in some sections (e.g., page 67) is inconsistent with Morris' dppc—clarify and standardize.

2. Quality of Evidence and Interpretation: GRADE rates evidence as low/very low, yet conclusions emphasize effectiveness (e.g., "thoracic techniques are more effective"). Temper claims to reflect this (e.g., "may be effective based on low-quality evidence"). Highlight implications for practice: given low evidence, these interventions should be adjunctive, not first-line. Address why thoracic techniques show larger effects— is it due to higher bias in those studies?

3. Publication Bias: Funnel plots suggest bias for disability (and pain in some versions), but Egger's/trim-and-fill do not. This discrepancy warrants discussion. The jackknife method identifies one influential study (Ho-Hee 2015) for disability—exclude it in sensitivity analysis and report if results change substantially.

4. Search Update: The search ends May 1, 2025, but the current date is September 24, 2025. While the revision addresses prior PLOS ONE concerns, perform a quick update to ensure no new studies (e.g., via PubMed alert). If none, state this. The inclusion of non-English studies (Korean, Chinese, Spanish) is a strength, but confirm translations were accurate.

5. Outcome Measures: Pain and disability are primary, but tools vary (VAS/NRS for pain, ODI/RMQ for disability). Discuss if this contributes to heterogeneity. Consider pooled results by specific tool in subgroups. Also, the abstract reports 36 studies in meta-analysis, but text specifies 31 for pain/24 for disability—clarify.

Minor Comments:

- Abstract: Update to reflect revised effect sizes (e.g., large for disability). Specify "thoracic manual techniques" for clarity.

- Methods: Search syntax in S1 Appendix is noted—ensure it's included. Define "thoracic techniques" more precisely (e.g., HVLA vs. mobilization).

- Results: Tables 2-4 are clear, but label effect sizes consistently (Morris' dppc vs. SMD in revisions). Figures 3-7 need legends for clarity.

- Discussion: Integrate recent studies (added in revision) more explicitly. Compare to Fabero-Garrido et al. (2024) on respiratory muscle training.

- References: Up-to-date, but check formatting (e.g., DOI consistency).

- Writing and Clarity: Minor typos (e.g., "Based on of the included studies" in conclusion; "dppc" vs. "SMD" inconsistencies). Ensure consistent terminology (e.g., "CLBP" vs. "LBP").

- Supplementary Materials: PRISMA checklist is referenced—verify completion. Include flow diagram (Fig 1) details.

**Do you want your identity to be public for this peer review?** For information about this choice, including consent withdrawal, please see our Privacy Policy

Reviewer #1: No

Reviewer #2: **Yes: ** Sarvenaz Karimi

---

## [Author Response · Author response to Decision Letter 1]

1 Oct 2025

Dear Editor and Reviewers,

We sincerely thank you for your thoughtful and constructive comments, which have greatly helped us improve the quality of our manuscript. We provide detailed, point-by-point responses to each reviewer’s comments and upload to the submission system.

With best regards

---

## [Decision Letter · Decision Letter 1]

20 Oct 2025

Dear Dr. Seyedhoseinpoor,

Thank you for submitting your manuscript to PLOS ONE. After careful consideration, we feel that it has merit but does not fully meet PLOS ONE’s publication criteria as it currently stands. Therefore, we invite you to submit a revised version of the manuscript that addresses the points raised during the review process.

We look forward to receiving your revised manuscript.

Kind regards,

Seyed Hamed Mousavi

Academic Editor

PLOS ONE

Journal Requirements:

Reviewers' comments:

Reviewer's Responses to Questions

**Comments to the Author**

Reviewer #1: All comments have been addressed

Reviewer #2: All comments have been addressed

2. Is the manuscript technically sound, and do the data support the conclusions?

Reviewer #1: Yes

Reviewer #2: Partly

3. Has the statistical analysis been performed appropriately and rigorously?

Reviewer #1: Yes

Reviewer #2: Yes

4. Have the authors made all data underlying the findings in their manuscript fully available?

Reviewer #1: Yes

Reviewer #2: Yes

5. Is the manuscript presented in an intelligible fashion and written in standard English?

Reviewer #1: Yes

Reviewer #2: Yes

Reviewer #1: Dear Authors

I would like to thank you for giving me an opportunity to consider this work for publication. You well done the a point by point answer to the comments of the reviewers Thanks

Reviewer #2: This revised manuscript shows substantial and meaningful improvement compared with the original version. The authors have carefully addressed most major reviewer comments raised in the first round, particularly those concerning methodological clarity, conceptual depth, and consistency in reporting. The study is now more coherent, transparent, and aligned with PRISMA and GRADE standards.

The topic remains highly relevant, as it investigates the effects of breathing exercises and thoracic manual techniques—an innovative, non-invasive approach to low back pain (LBP) rehabilitation. The revision strengthens the overall scientific and clinical credibility of the work. However, a few issues related to residual heterogeneity, limited exploration of moderators, and overinterpretation of low-certainty evidence still need refinement before acceptance.

Minor Comments

Include abbreviations in table footnotes for clarity (e.g., RMQ = Roland–Morris Questionnaire, ODI = Oswestry Disability Index).

Ensure figure legends clearly describe the direction of effect and metric (Morris’ dppc).

Confirm that all supplementary materials (search syntax, PRISMA checklist, flow diagram) are included and updated.

Check reference formatting (consistent DOI presentation)

**Do you want your identity to be public for this peer review?** For information about this choice, including consent withdrawal, please see our Privacy Policy

Reviewer #1: No

Reviewer #2: **Yes: ** Sarvenaz Karimi-ghasemabad

---

## [Author Response · Author response to Decision Letter 2]

21 Oct 2025

Dear respected Editor and reviewers,

We are grateful for the reviewers’ valuable feedback, which has helped us significantly strengthen the scientific rigor and clarity of our manuscript. We believe these revisions further enhance the transparency, methodological robustness, and clinical relevance of the study.

Regards

Reviewer #1

Comment:

Dear Authors,

I would like to thank you for giving me an opportunity to consider this work for publication. You well done the point-by-point answer to the comments of the reviewers. Thanks.

Response:

We sincerely thank the reviewer for their kind words and for taking the time to evaluate our revised manuscript. We greatly appreciate your recognition of our efforts to address all prior comments thoroughly and improve the clarity and quality of the paper.

Reviewer #2

General Comment:

This revised manuscript shows substantial and meaningful improvement compared with the original version. The authors have carefully addressed most major reviewer comments raised in the first round, particularly those concerning methodological clarity, conceptual depth, and consistency in reporting. The study is now more coherent, transparent, and aligned with PRISMA and GRADE standards.

Response:

We thank the reviewer for the positive and constructive feedback. We are pleased that you found the revised version substantially improved in terms of methodological clarity, conceptual depth, and consistency with PRISMA and GRADE standards. We have carefully addressed the remaining concerns as detailed below.

General Comment:

The topic remains highly relevant, as it investigates the effects of breathing exercises and thoracic manual techniques—an innovative, non-invasive approach to low back pain (LBP) rehabilitation. The revision strengthens the overall scientific and clinical credibility of the work. However, a few issues related to residual heterogeneity, limited exploration of moderators, and overinterpretation of low-certainty evidence still need refinement before acceptance.

Response:

We appreciate the reviewer’s insightful comment. The Discussion and Conclusion sections have been revised to more clearly address residual heterogeneity, acknowledge potential moderators, and moderate the interpretation of low-certainty evidence. These changes ensure a more balanced and transparent presentation of our findings.

Specific Comments and Responses

1. Comment: Include abbreviations in table footnotes for clarity (e.g., RMQ = Roland–Morris Questionnaire, ODI = Oswestry Disability Index).

Response: We thank the reviewer for the observation. The abbreviations (e.g., RMQ, ODI, etc.) were already included in the table footnotes in the previous version; we have rechecked and ensured that all abbreviations are clearly defined for consistency and clarity.

2. Comment: Ensure figure legends clearly describe the direction of effect and metric (Morris’ dppc).

Response: Figure legends have been revised to explicitly indicate the direction of effect and the metric used (Morris’ dppc).

3. Comment: Confirm that all supplementary materials (search syntax, PRISMA checklist, flow diagram) are included and updated.

Response: We confirm that all supplementary materials have been included and updated, including the full search syntax, PRISMA checklist, and revised flow diagram reflecting the final number of included studies.

4. Comment: Check reference formatting (consistent DOI presentation).

Response: We have reviewed and corrected all references to ensure consistent DOI formatting according to the journal’s style guidelines.

---

## [Decision Letter · Decision Letter 2]

2 Dec 2025

Effects of Breathing Exercise and Thoracic Techniques on Pain and Disability in Low Back Pain: A Systematic Review and Meta-analysis

PONE-D-25-26341R2

Dear Dr. Seyedhoseinpoor,

We’re pleased to inform you that your manuscript has been judged scientifically suitable for publication and will be formally accepted for publication once it meets all outstanding technical requirements.

**Reviewer2 raised a few minor comments that can be addressed during the proofing stage.**

Kind regards,

Seyed Hamed Mousavi

Academic Editor

PLOS ONE

Additional Editor Comments (optional):

Reviewers' comments:

Reviewer's Responses to Questions

**Comments to the Author**

Reviewer #2: All comments have been addressed

2. Is the manuscript technically sound, and do the data support the conclusions?

Reviewer #2: Yes

3. Has the statistical analysis been performed appropriately and rigorously?

Reviewer #2: Yes

4. Have the authors made all data underlying the findings in their manuscript fully available?

Reviewer #2: Yes

5. Is the manuscript presented in an intelligible fashion and written in standard English?

Reviewer #2: (No Response)

Reviewer #2: PLOS ONE Peer-Review Report

Manuscript Number: PONE-D-25-26341R2

Title: Effects of Breathing Exercise and Thoracic Techniques on Pain and Disability in Low Back Pain: A Systematic Review and Meta-analysis

Overall Recommendation: ACCEPT with very minor revisions

(The manuscript is now of publishable quality for PLOS ONE, and the authors have satisfactorily addressed almost all previous concerns.)

Summary

This R2 version represents a clear and substantial improvement over the original submission and R1. The authors have carefully responded to both reviewers, updated supplementary files, clarified statistical methods, improved figure legends, ensured consistent abbreviation definitions, and moderated the language in the Conclusion to better reflect the low/very low certainty of evidence according to GRADE.

The topic remains highly clinically relevant: combining breathing/retraining of the diaphragm with thoracic manual therapy is an emerging, low-risk approach in non-specific low back pain, and this is the most comprehensive meta-analysis on this specific combination to date .

Strengths

• Comprehensive, pre-registered (PROSPERO), PRISMA-compliant search with no language restrictions and translation of non-English trials

• Appropriate use of Morris’ dppc2 (pre-post-control effect size) for the predominant study designs

• Transparent risk-of-bias assessment (RoB 2 + robvis figures) and correct application of GRADE (downgraded for inconsistency, imprecision, and risk of bias)

• Subgroup and sensitivity analyses that partially explain the high heterogeneity

• Clinically useful distinction between “breathing exercises only” vs. “thoracic manual techniques” (the latter showing clearly superior effects)

• Balanced discussion of mechanisms (diaphragm–core synergy, thoracic stiffness, regional interdependence)

Remaining very minor issues (can be corrected in proof stage)

1. Abstract (line about heterogeneity): the sentence “However, statistical heterogeneity remained across studies” is correct but slightly understates the issue. Consider changing to: "However, substantial statistical heterogeneity (I² > 85%) persisted in most analyses.”

2. Table 1 (Characteristics of included studies): Please add one column indicating “LBP duration” (acute/subacute/chronic/mixed) for each study – this is a major source of clinical heterogeneity and would take only a few minutes to insert.

3. Figure legends (Forest plots): Already much improved, but add one short sentence to each: Negative values favour the thoracic/breathing intervention.”

4. Discussion – page ~29: Add one small typo: “catious” → “cautious”

Final decision

The manuscript is now suitable for publication in PLOS ONE.

Decision: ACCEPT with very minor revisions

(The above four points can be addressed by the editorial office during copy-editing or by the authors in proof. No further peer-review round is required.)

Congratulations to the authors on a well-conducted and clinically meaningful systematic review.

**Do you want your identity to be public for this peer review?** For information about this choice, including consent withdrawal, please see our Privacy Policy

Reviewer #2: **Yes: ** Sarvenaaz Karimi-ghasemabad

---

## [Editor Report · Acceptance letter]

PONE-D-25-26341R2

PLOS One

Dear Dr. Seyedhoseinpoor,

I'm pleased to inform you that your manuscript has been deemed suitable for publication in PLOS One. Congratulations! Your manuscript is now being handed over to our production team.

Kind regards,

on behalf of

Dr. Seyed Hamed Mousavi

Academic Editor

PLOS One